# Hydrophobicity/Oleophilicity of Autoclaved Aerated Concrete (AAC) Grains Coated with Oleic and Stearic Acids for Application as Oil/Water Separating Filtration and Adsorbent Materials in Vietnam

**Akihiro Matsuno [1,*] and Ken Kawamoto [1,2]**

[1]  Graduate School of Science and Engineering, Saitama University,
    255 Shimo-okubo, Sakura-ku, Saitama 3388570, Japan
[2]  Innovative Solid Waste Solutions (Waso), Hanoi University of Civil Engineering,
    No. 55 Giai Phong Street, Hai Ba Trung District, Hanoi 11616, Vietnam
*   Correspondence: matsuno2017@mail.saitama-u.ac.jp; Tel.: +81-488299159

**Abstract:** Separation of oil and water is an effective technique to treat oily wastewater. For examining the applicability of porous grains coated with hydrophobic agents (HA) as low-cost and easily available filtration and adsorbent materials in the separation of oil and water, this study assessed the hydrophobicity/oleophilicity of porous grains made from autoclaved aerated concrete scrap coated with low-cost and harmless hydrophobic agents such as oleic and stearic acids. Tests using a sessile droplet method showed unique relationships between the contact angles (CA) of water droplets in air ($CA_{wa}$), oil droplets in water ($CA_{ow}$) and coated HA concentrations. The $CA_{wa}$ increased linearly with increasing HA concentration and then became almost constant and/or gently increased after a specific point, indicating that a minimum coating amount gives the maximum hydrophobicity to HA-coated porous grains exists. The $CA_{ow}$ gradually decreased exponentially with increasing HA concentration. In particular, the $CA_{ow}$ of porous grains coated with stearic acid decreased with increasing of HA concentration in a two-step process. Furthermore, analyses of the Pearson correlation showed that both $CA_{wa}$ and $CA_{ow}$ correlated well with the specific surface area (SSA), implying that the SSA is a good indicator as a quick assessment of hydrophobicity/oleophilicity of HA-coated porous grains.

**Keywords:** hydrophobicity/oleophilicity; autoclaved aerated concrete; contact angle; sessile droplet method; oleic acid; stearic acid; oil/water separation; filtration and adsorbent materials; Vietnam

## 1. Introduction

Oil and grease in water are major pollutants of the water environment. The improper discharge and treatment of oily wastewater cause significant environmental pollution of surface and groundwater, rivers and ponds, and seawater. Especially, developing countries with rapid urbanization, industrialization, and population growth face serious water pollution by oily wastewater due to the insufficient development of wastewater treatment systems, resulting in severe impacts on human health and natural ecosystems (e.g., [1–3]). For example, Vietnam (one of the growing developing countries) reported that the inadequate treatment and improper disposal of oily wastewater as well as oil spills generated by human activities and industries such as petrochemicals, food processing, textiles, metals, mining, biopharmaceuticals, and oil and gas refineries cause serious water pollution in the whole country [4–6], and urgent actions are necessary to conserve a sound water environment and sustainable development and to prevent economic losses from improper oily wastewater treatment [1–3].

Nowadays, many technologies such as flotation, coagulation, chemical treatment, gravity separation, biological treatment, and oil/water separation have been developed and applied to treat oily wastewater [7,8]. Among of them oil/water separation technologies using hydrophobic and oleophilic membranes/meshes/grains as filtration and adsorbent materials have been intensively developed due to their high treatment performance and cost effectiveness (e.g., [9,10]). Because the oil/water separation technologies do not need a centralized wastewater collection system and a large volume of water treatment tanks/ponds, they can be incorporated into small-scale decentralized wastewater treatment systems such as fixed-bed filtration tanks targeting domestic wastewater in rural areas (with a small number of households) and wastewater from food processing factories and craft villages [11–13]. From the viewpoint of durability and feasibility, furthermore, the filtration and/or adsorbent materials used in the oil/water separation made from easily available and environmentally sound recycled materials are desirable, particularly in Vietnam [11–13].

One of the most important characteristics to develop a high-performance oil/water separation system is the hydrophobicity/oleophilicity of the membranes/meshes/grains used as filtration and/or adsorbent materials [14–20]. The degree of hydrophobicity/oleophilicity is commonly assessed by measuring contact angles (CA) in air and/or water [21–24]. In a recent study [25], polyvinylidene fluoride (PVDF) membranes were coated with a suspension of waste brick powder (WBP) and then with sodium alginate by vacuum filtration to produce WBP-coated films. Shi et al. showed that WBP-coated membranes exhibited excellent underwater superoleophobicity, including under corrosive conditions and low adhesion to crude oil [25]. In addition, the WBP-coated membrane can separate crude oil-in-water emulsions with high separation efficiencies and permeating flux. Li et al. (2020a) showed that a $TiO_2$ coating applied to WBP [26] produced samples with superhydrophobic/superoleophilic properties [27], as dud Li et al. (2020b) using ZnO coating. Li et al. (2020a) and Li et al. (2020b) demonstrated that filter layers with opposite wettability exhibit high separation efficiency and flux [26,27]. On the other hand, Zhao et al. showed that a layer of waste peanut shells caused super-oleophobicity (oil contact angle of 150 ± 5) under water due to separation of immiscible oil/water mixtures by a hydrophilic substance and exhibited a desirable flux rate (2636.9 L $m^{-2} \cdot h^{-1}$) and high separation efficiency (>99.5%) for a surfactant stabilized water-in-oil emulsion [28].

As described above, many studies to assess the surface hydrophobicity/oleophilicity of filtration and/or adsorbent materials in the oil/water separation were based on the CA measurements. However, the CA measurements of hydrophobized/oleophilized surfaces in both oily water–air and oil–water systems (i.e., oil contact angles in air and water) have not been fully investigated, even though the oil contact angles in both air and water are needed to characterize the degree of hydrophobicity/oleophilicity of filtration and adsorbent materials and to assess the oil/water separation performance under both continuous (oil-in-water) and discontinuous (oily water in air) water flow conditions. Moreover, most previous studies that use hydrophobized grains for the oil/water separating materials used non-porous gains with a small specific surface area, such as quartz sands, and a limited number of studies have examined the applicability of porous grains (rich in fine pores inside and with a large specific surface area) as oil/water separating materials [29].

To encourage green practices and sustainable development, Vietnam is encouraging all economic sectors to invest in non-baked building material production and trading [e.g., 30]. Among the non-baked building materials, autoclaved aerated concrete (AAC) is readily manufactured in Vietnam and widely used in many building applications [30]. AAC has a unique pore structure that consists of inner-pores (μm to nm scale) and inter-pores (mm to μm scale) [31–35]. The scrap and waste AAC has been increasing, but most AAC scrap is currently stored/dumped without reuse or recycling in many factories. Although the reuse and recycling of AAC scrap are highly desirable for sustainable

consumption and production to implement the circular economy in Vietnam [36,37], till now very limited research has been done on the utilization of AAC scrap to reduce the amount that is dumped [38]. Thus, research is needed to promote development utilizing the unique porous characteristics of AAC.

Therefore, to promote low cost and high-performance oil/water separating filtration and adsorbent materials utilizing porous grains made from AAC scrap in Vietnam (VN-AAC), this research assessed the hydrophobicity (oleophilicity) of AAC grains coated with hydrophobic agents harmless to humans and the environment, such as oleic and stearic acids [39]. The specific objectives were (1) to measure the CA of water droplets in air ($CA_{wa}$) and the CA of oil droplets in water ($CA_{ow}$) of hydrophobized AAC grains in three grain sizes (0.106–0.250, 0.250–0.850, and 0.850–2.00 mm) and assess the hydrophobicity (oleophilicity) by examining the measured $CA_{wa}$ and $CA_{ow}$ values as a function of the amount of coated hydrophobic agents, and (2) to examine the correlations of measured $CA_{wa}$ and $CA_{ow}$ with physicochemical parameters such as specific surface area and organic carbon content. For comparing the tested results from AAC grains to those from non-porous grains, commercially available sands (grain sizes of 0.18–2.00 and 0.30–2.00 mm) used for sand filtration systems were also used in this research.

## 2. Materials and Methods

### 2.1. AAC Grains and Sands

Block-like AAC scrap (approximately 500 kg) was taken from Viglacera Joint Stock Company in Bac Ninh Province, Vietnam (21°11′50.8″ N, 106°00′42.8″ E) [40]. AAC is a typical lightweight cementitious material with a robust skeleton and pore structure (porous medium) and is widely used in many architectural applications worldwide due to its low density, thermal conductivity, and high heat resistance [41,42]. AAC is known to form a calcium silicate hydrate, tobermorite [$Ca_5Si_6(O, OH, F)_{18}5H_2O$], during the manufacturing process [42,43]. The block-like AAC scrap was first crushed to less than 10 mm and sieved in the laboratory to make AAC grains. Then, the AAC grains were washed gently with a low-foaming neutral detergent and rinsed thoroughly with distilled water. After being rinsed, the AAC grains were air dried and sieved again into three grain fractions, 0.106–0.250 mm (fine), 0.250–0.850 mm (middle), and 0.850–2.00 mm (coarse) [44]. For comparison of tested data from AAC grains, two commercially available sands (Nihon Genryo Materials Co., Ltd., Kanagawa, Japan) that meet the Japanese standards of filtration [45] were also used in this research. Sand with the size of 0.18–2.00 mm was used for slow filtration, and sand with the size of 0.30–2.00 mm was used for rapid filtration. The AAC grains and sands tested in this research are shown in Figure 1.

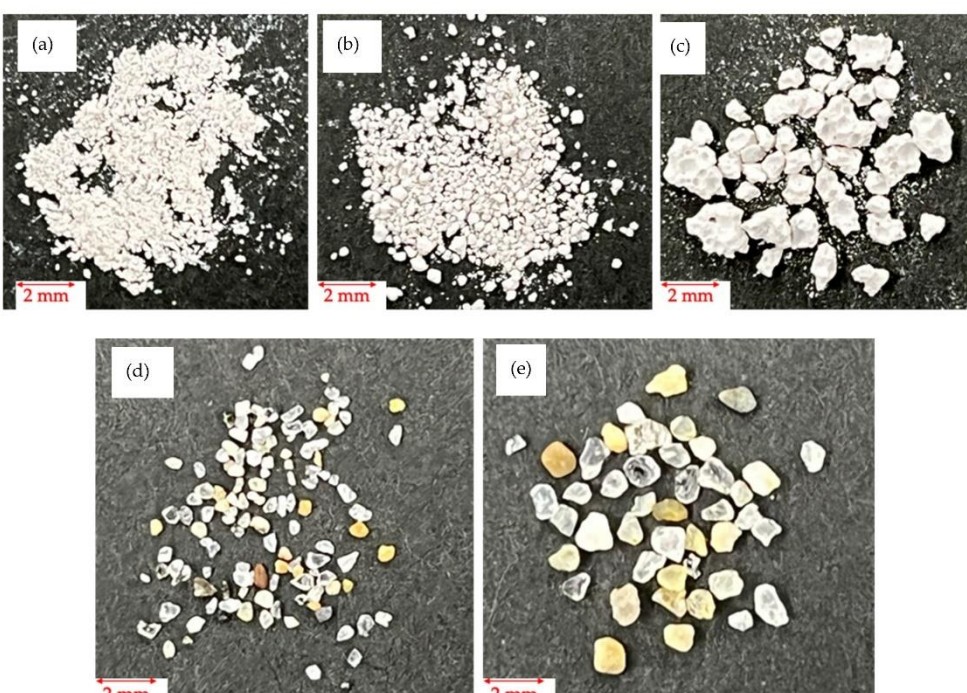

**Figure 1.** Samples: (**a**) VN-AAC (Grain size: 0.106–0.25 mm); (**b**) VN-AAC (Grain size: 0.25–0.85 mm); (**c**) VN-AAC (Grain size: 0.85–2.00 mm) (**d**) Sand (Grain size: 0.18–2.00 mm); (**e**) Sand (Grain size: 0.30–2.00 mm).

### 2.2. Oil

In this study, soybean oil with a high percentage of linoleic acid (53.5%) and linolenic acid (6.9%) (Wako 1st grade; Fujifilm Wako Pure Chemical Corp., Tokyo, Japan) was used. Soybean oil is a commonly consumed domestic oil in Vietnam [46] and is liquid at room temperature.

### 2.3. Hydrophobic Agents and Coating

Two types of hydrophobic agents (HA), stearic and oleic acids (SA and OA), were chosen to hydrophobically (oleophilically) coat the grains tested in this research [39,47–49]. Both OA and SA are harmless to humans and the environment. Olive oil is known to be rich in OA, and SA is plentiful in beef tallow and other animal and vegetable fats, materials readily available at low cost in Vietnam. In the laboratory, commercially available OA (molar mass: 282.46 g/mol, density: 0.895 g/cm³) (Kanto Chemical Corp., Tokyo, Japan) and SA (molar mass: 284.47 g/mol, density: 0.940 g/cm³) (Fujifilm Wako Pure Chemical Corp., Tokyo, Japan) were used. OA is a mono-unsaturated omega-9 fatty acid with the chemical formula $CH_3(CH_2)_7CH:CH(CH_2)_7COOH$, and SA is a saturated fatty acid with the chemical formula $CH_3(CH_2)_{16}COOH$.

The hydrophobic coating was done as follows: air-dried AAC grains and sands were soaked in a beaker along with the solvent to reach target HA concentrations (g/kg) [= $M_s/M_d$, where $M_s$ is the dry mass of HA (g) and $M_d$ is the dry mass of tested grains (kg)]. It is noted that at average room temperature, SA is a solid (powder) (melting point: 69 °C), and OA is a liquid (melting point 13–16 °C). Therefore, considering the polarities of SA and OA, diethyl ether and ethanol were used as solvents for SA and OA, respectively. The SA and diethyl ether were thoroughly mixed using an electronic mixer before adding the sample. After volatilization of the solvent, the coated samples were stored for at least 48 h in a climate-controlled lab at 20 °C and 60% humidity to equilibrate and stored in a plastic bag after air-drying [50].

### 2.4. Measurement of Contact Angles

Contact angles of water droplets in air ($CA_{wa}$) for tested grains were measured by the sessile droplet method (SDM) [47,50,51] using a digital microscope (VHX-900 series, KEYENCE Japan, Tokyo, Japan). The grains were put on a smooth glass slide to which double-sided tape was attached. The glass slide was then lightly tapped to remove the excess sample, the glass slide was placed on the microscope camera stage, and a small drop of distilled water ($50 \pm 5$ µL) was dropped onto the grains using a syringe. Horizontal microscope images were recorded to determine $CA_{wa}$ with time, as shown in Figure 2a.

The contact angles of an oil droplet in water ($CA_{ow}$) were measured following the previous literature [52–54]: the test grains were placed on a smooth glass slide using a piece of double-sided tape. Then the glass slide was attached to a glass rod, and the glass rod was placed in a 2 L beaker filled with distilled water. A small drop (0.1 mL) of soybean oil was dripped from a microsyringe with the needle bent and pointing upward toward the sample attached to the underside of the slide glass. The $CA_{ow}$ between the oil droplet and the grains is exemplified in Figure 2b.

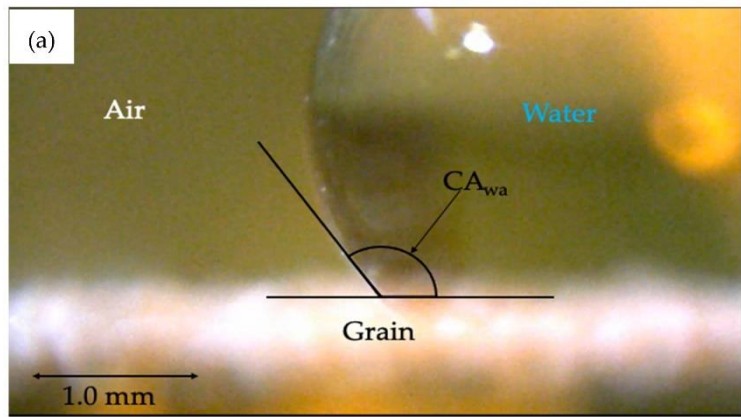

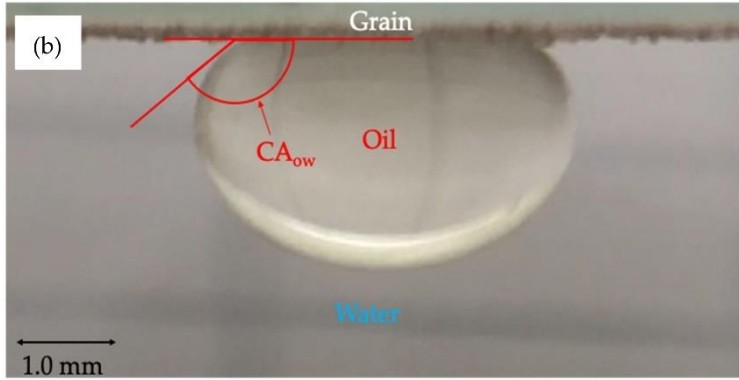

**Figure 2.** Sessile droplet method to measure contact angles: (**a**) Water in air ($CA_{wa}$); (**b**) Oil in water ($CA_{ow}$).

### 2.5. Physical and Chemical Properties of Tested Grains and Characterization of Hydrophobicity of HA-Coated Grains

Basic physical and chemical properties such as specific gravity ($G_s$), loss on ignition (LOI), moisture content in the air-dried condition ($w_{AD}$), pH, and electrical conductivity (EC) were measured [55,56]. The organic carbon (OC) content of HA-coated grains was measured by an elemental analyzer (FLASH 2000, Thermo Fisher Scientific, Inc., Waltham, MA, USA). The measured OC was also used to examine the relationships between added (mixed) HA and calculated HA based on OC (see Figure A1 in Appendix

A). To characterize the surface hydrophobicity (oleophilicity) of tested grains, the A/B ratio was calculated based on the spectra measured by a Fourier transform infrared (FT-IR) spectrometer (Tensor II, Bruker Ltd., Banner Lane, UK). The FT-IR spectra focused on two absorption bands that primarily represent hydrophobic (C-H group) and hydrophilic (C=O group) functional groups. For hydrophobic methyl and methylene groups, the C-H bands occur at 2920 $cm^{-1}$ (asymmetric stretch) and 2860 $cm^{-1}$ (symmetric stretch) [57]. Here, both bands are combined into one band (3020–2800 $cm^{-1}$), which we designated as absorption band A. Hydrophilic C=O groups occur at 1640–1615 and 1740–1720 $cm^{-1}$ [58,59]. Here, we used slightly different bands, 1640 to 1620 and 1740 to 1710 $cm^{-1}$, to exclude a possible overlap with C=C and amide bands and denoted both as absorption band B. The OH bands were not considered because they could reflect differences in water content. From this, the ratio of hydrophobicity to hydrophilicity (A/B ratio) was determined [60].

The surface properties of the coated/non-coated grains were characterized by measuring the Brunauer–Emmett–Teller (BET) specific surface area (SSA in $m^2/g$) using a porosity analyzer (TriStar II, Micromeritics Instruments Corp., Norcross, GA, USA) [61]. The total pore volume ($V_T$) of non-coated grains was also measured using a porosity analyzer based on the Barrett–Joyner–Halenda method [62]. In addition, a scanning electron microscope (SEM) (TM4000plus, Hitachi High-Technologies Corp., Tokyo, Japan) and an energy dispersive X–ray spectroscope (EDS) (AZtecOne, Oxford Instruments, Abingdon, Oxon, UK) were used to investigate the morphological features and chemical compositions of HA-coated grains.

## 3. Results and Discussion

### 3.1. Physical and Chemical Characterization

Table 1 shows the physical and chemical properties of tested samples. The values of the uniformity coefficient ($U_c = D_{50}/D_{10}$) of tested samples ranged from 0.83 to 1.64, indicating that the samples were "uniform" [63]. For VN-AAC samples, the measured specific surface area (SSA) decreased with increasing grain size. The total pore volume ($V_T$), loss on ignition (LOI), and air-dried water content ($w_{AD}$) of VN-AAC, on the other hand, did not vary with grain size, and those values were one order of magnitude greater than those of sand. In particular, the high values of SSA and $V_T$ of VN-AAC indicated that these samples were porous. VN-AAC samples were alkaline in water (pH = 8.6~9.0) and neutral in KCl solution (pH = 7.2–7.3). SEM-EDS images of OA- and SA-coated VN-AAC grains are exemplified in Figure 3. It can be seen the carbon (C) originating from coated HA materials (OA and SA) covered the tested grain surfaces well.

**Table 1.** Basic physical and chemical properties of tested samples in this study.

| Sample | Grain Size (mm) | $D_{10}$ (mm) | $D_{50}$ (mm) | $D_{60}$ (mm) | $U_c$ ($D_{50}/D_{10}$) | SSA ($m^2/g$) | $V_T$ ($cm^3/g$) | LOI (%) | $w_{AD}$ (%) | $G_s$ ($g/cm^3$) | EC (mS/cm) | pH ($H_2O$) | pH (1 mol KCl) |
|---|---|---|---|---|---|---|---|---|---|---|---|---|---|
| VN-AAC | 0.106–0.25 | 0.15 | 0.20 | 0.21 | 1.44 | 15.3 | $4.5 \times 10^{-2}$ | 9.2 | 1.44 | 2.57 | 1.04 | 8.90 | 7.27 |
| VN-AAC | 0.25–0.85 | 0.34 | 0.51 | 0.56 | 1.64 | 16.9 | $4.9 \times 10^{-2}$ | 9.5 | 2.03 | 2.55 | 0.94 | 8.57 | 7.20 |
| VN-AAC | 0.85–2.00 | 0.90 | 1.20 | 1.30 | 0.83 | 17.1 | $4.8 \times 10^{-2}$ | 9.4 | 2.01 | 2.46 | 0.83 | 8.97 | 7.33 |
| Slow sand | 0.18–2.00 | 0.31 | 0.48 | 0.55 | 1.54 | 3.0 | $6.0 \times 10^{-3}$ | 0.6 | 0.44 | 2.66 | 0.01 | 7.83 | 6.13 |
| Rapid sand | 0.30–2.00 | 0.62 | 0.79 | 0.83 | 1.28 | 3.5 | $7.0 \times 10^{-3}$ | 0.5 | 0.40 | 2.63 | 0.02 | 7.27 | 6.00 |

$D_{10}$: Grain size at 10% passing, $D_{50}$: Grain size at 50% passing (Median diameter), $D_{60}$: Grain size at 60% passing, $U_c$: Uniformity coefficient, SSA: Specific surface area, $G_s$: Specific gravity, $w_{AD}$: Air-dried water content, EC: Electrical conductivity, pH: Potential hydrogen was measured by using distilled water and 1 mol KCl solution (S:L = 1:2.5).

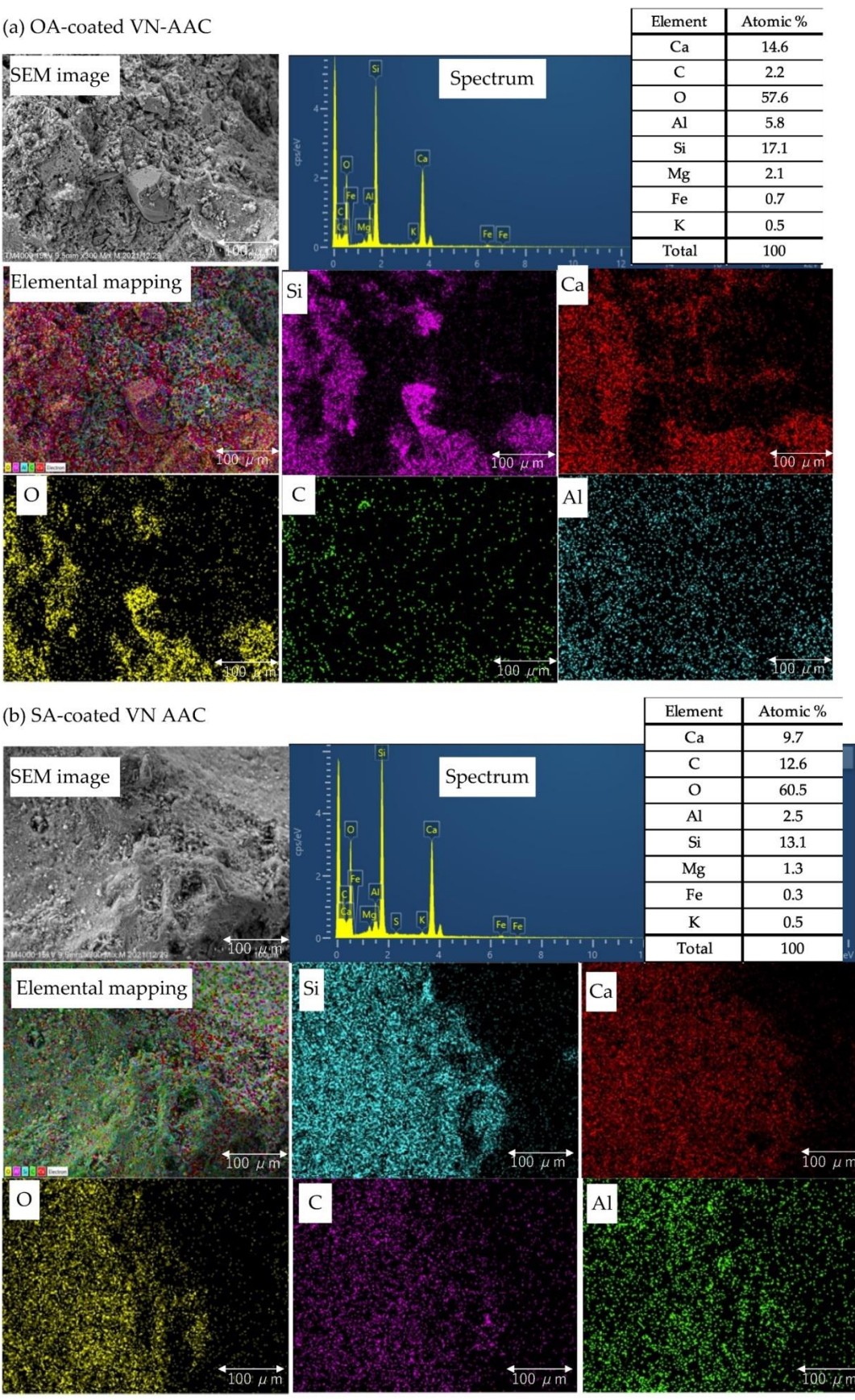

**Figure 3.** SEM-EDS images of (**a**) OA-coated VN-AAC grain (0.85–2.00 mm) at HA concentration = 14 g/kg and (**b**) SA-coated VN-AAC grain (0.85–2.00 mm) at HA concentration = 42 g/kg.

*3.2. Contact Angles of Water in Air (CA$_{wa}$)*

　　Measured contact angles of water droplets in air (CA$_{wa}$) of tested samples as a function of the HA coating concentration are shown in Figure 4. For both OA- and SA-coated VN-AAC samples, the CA$_{wa}$ values increased linearly with increasing HA concentration and reached 98–140°, and then became almost constant (Figure 4a,b,f) and/or gently increased (Figure 4c,g,h) after a specific point (i.e., point of intersection, hereafter labeled "PoI" and the contact angle at PoI labeled "CA$_{wa,PoI}$"). The slopes in the linear increment section (ΔS$_1$) and those in the gentle increment section (ΔS$_2$) were calculated and are given in the figure. As well, the HA concentrations corresponding to the PoI (HA$_{wa,PoI}$) and the maximum contact angles of a water droplet in air in the measured range (CA$_{wa,max}$) were read and are summarized in Table 2 For VN-AAC coated with both OA and SA, both CA$_{wa,max}$ and CA$_{wa,PoI}$ decreased with increasing grain size, indicating that a finer grain gave a higher hydrophobicity.

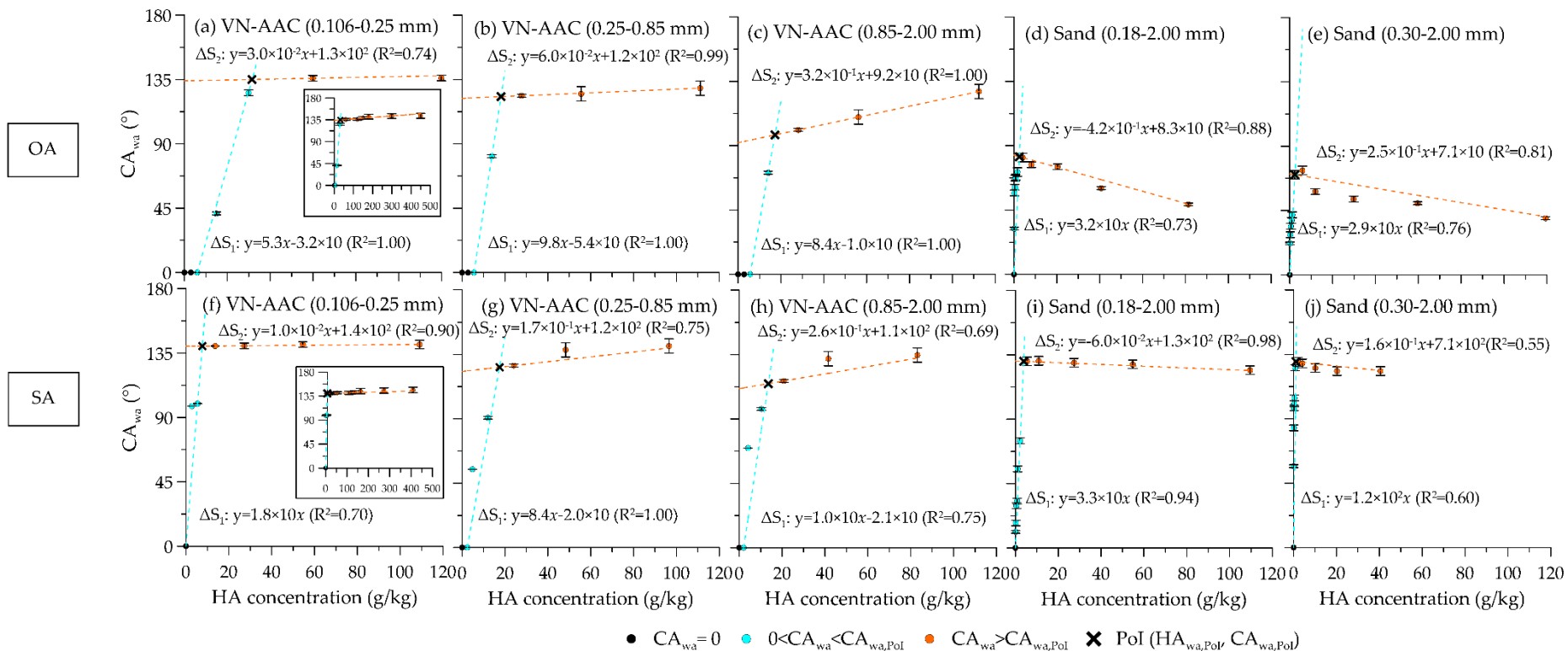

**Figure 4.** Relationships between CA$_{wa}$ and HA concentrations of OA- and SA-coated VN-AAC grains and sands. OA-coated VN-AAC grains: (**a**) Grain size = 0.106–0.25 mm, (**b**) Grain size = 0.25–0.85 mm, and (**c**) Grain size = 0.85–2.00 mm. OA-coated sands: (**d**) Grain size = 0.18–2.00 mm and (**e**) Grain size = 0.30–2.00 mm. SA-coated VN-AAC grains: (**f**) Grain size = 0.106–0.25 mm, (**g**) Grain size = 0.25–0.85 mm, and (**h**) Grain size = 0.85–2.00 mm. SA-coated sands: (**i**) Grain size = 0.18–2.00 mm, and (**j**) Grain size = 0.30–2.00 mm.

**Table 2.** Summary of hydrophobicity/oleophilic parameters in this study and the literature.

| Sample | Grain Size (mm) | Coating | CA$_{wa}$ | | | | | CA$_{ow}$ | | | Ref. |
|---|---|---|---|---|---|---|---|---|---|---|---|
| | | | CA$_{wa,max}$ (°) | CA$_{wa,PoI}$ (°) | HA$_{wa,PoI}$ (g/kg) | ΔS$_1$ (Slope) | ΔS$_2$ (Slope) | CA$_{ow,max}$ (°) | CA$_{ow,PoI}$ (°) | HA$_{ow,PoI}$ (g/kg) | |
| VN-AAC | 0.106–0.25 | | 143 | 135 | 31.5 | 5.28 | 0.03 | 140 | 131 | 30.0 | |
| VN-AAC | 0.25–0.85 | OA | 129 | 123 | 18.2 | 9.77 | 0.06 | 130 | 124 | 13.9 | |
| VN-AAC | 0.85–2.00 | | 128 | 98 | 17.2 | 8.42 | 0.32 | 124 | 119 | 28.1 | |
| Slow sand | 0.18–2.00 | | 82 | 82 | 2.5 | 32.36 | −0.42 | 138 | 0 | 20.3 | |
| Rapid sand | 0.30–2.00 | | 73 | 70 | 2.4 | 28.77 | −0.25 | 137 | 0 | 29.9 | This Study |
| VN-AAC | 0.106–0.25 | | 146 | 140 | 7.7 | 18.29 | 0.01 | 138 | 80 | 54.5 | |
| VN-AAC | 0.25–0.85 | SA | 141 | 126 | 17.5 | 8.37 | 0.17 | 130 | 121 | 12.2 | |
| VN-AAC | 0.85–2.00 | | 135 | 115 | 13.6 | 9.99 | 0.26 | 124 | 106 | 20.8 | |
| Slow sand | 0.18–2.00 | | 131 | 131 | 3.9 | 33.49 | −0.06 | 138 | 0 | 27.4 | |
| Rapid sand | 0.30–2.00 | | 130 | 130 | 1.1 | 115.14 | −0.16 | 142 | 0 | 10.1 | |
| Glass bead | 0.075–0.25 | | 106 | 106 | 0.3 | 172.00 | 22.5 | NA | NA | NA | |
| Accusand | 0.105–0.25 | | 76 | 76 | 1.0 | 58.00 | 15.1 | NA | NA | NA | |
| Toyoura sand | 0.105–0.25 | OA | 97 | 97 | 0.3 | 388.00 | 0.25 | NA | NA | NA | Wijewardana et al., 2015 [47] |
| Narita sand | 0.105–0.25 | | 94 | 94 | 1.3 | 86.00 | 0.25 | NA | NA | NA | |
| Narita sand | 0.25–0.425 | | 93 | 93 | 1.0 | 93.00 | 0.1 | NA | NA | NA | |
| Narita sand | 0.425–0.84 | | 79 | 79 | 1.3 | 89.00 | 0.11 | NA | NA | NA | |
| Toyoura sand | 0.105–0.25 | OA | 101 | 101 | 0.3 | 329.00 | 0.25 | NA | NA | NA | Subedi et al., 2012 [39] |
| Quartz sand | 0.05–0.25 | | 99 | 99 | 0.3 | 329.00 | 0.25 | NA | NA | NA | González-Peñaloza et al., 2013 [64] |
| Quartz sand | 0.25–0.5 | SA | 100 | 100 | 1.0 | 36.00 | 0.06 | NA | NA | NA | |
| Quartz sand | 0.5–2.0 | | 101 | 101 | 2.0 | 28.00 | 0.01 | NA | NA | NA | |

For both OA- and SA-coated sands, on the other hand, the CA$_{wa}$ values increased rapidly, reached a peak, and decreased linearly with increasing HA concentration (Figure 4d,e,i,j; Table 2). The CA$_{wa,max}$ values of sands were smaller than those of VN-AAC (73–82° for OA-coated and 130–131° for SA-coated). In particular, CA$_{wa,max}$ = 73–82° of OA-coated slow sand means that the tested sand does not show hydrophobicity in air (non-water repellency).

The measured CA$_{wa,max}$, CA$_{wa,PoI}$, HA$_{wa,PoI}$, ΔS$_1$, and ΔS$_2$ in this study were compared to values of sands and glass beads reported by previous studies [39,47,64] (Table 2). Basically, the values of OA- and SA-coated sands in this study were similar to those in the previous studies and the difference in CA$_{wa,max}$ was within 10° [39,47,64]. Again, it can be found that the OA- and SA-coated AAC grains in this study gave high hydrophobicity with high values of CA$_{wa,max}$ and CA$_{wa,PoI}$.

### 3.3. Contact Angles of Oil in Water (CA$_{ow}$)

Measured contact angles of an oil droplet in water (CA$_{ow}$) of tested samples as a function of HA coating concentration are shown in Figure 5. For OA-coated VN-AAC samples (Figure 5a–c), the CA$_{ow}$ values gradually decreased with increasing HA

concentration (i.e., $CA_{ow,max}$ appeared at HA concentration = 0 g/kg). For SA-coated VN-AAC samples, on the other hand, the $CA_{ow}$ decreased with increasing HA concentration in two steps, and a clear PoI (i.e., $HA_{ow,PoI}$) can be found. The measured values were fitted by an exponential equation in each step:

$$\text{Step 1: } y = a \exp(-bx) \tag{1}$$

$$\text{Step 2: } y = a' \exp[-b'(x - c)] \tag{2}$$

where a is the $CA_{ow,max}$ at HA concentration = 0 g/kg (°), b is the fitting parameter in Step 1, a′ is the $CA_{ow,PoI}$ at $HA_{ow,PoI}$ (°), b′ is the fitting parameter in Step 2, and c is the $HA_{ow,PoI}$ (g/kg). The measured $CA_{ow,max}$, $CA_{ow,PoI}$, and $HA_{ow,PoI}$ are also shown in Table 2.

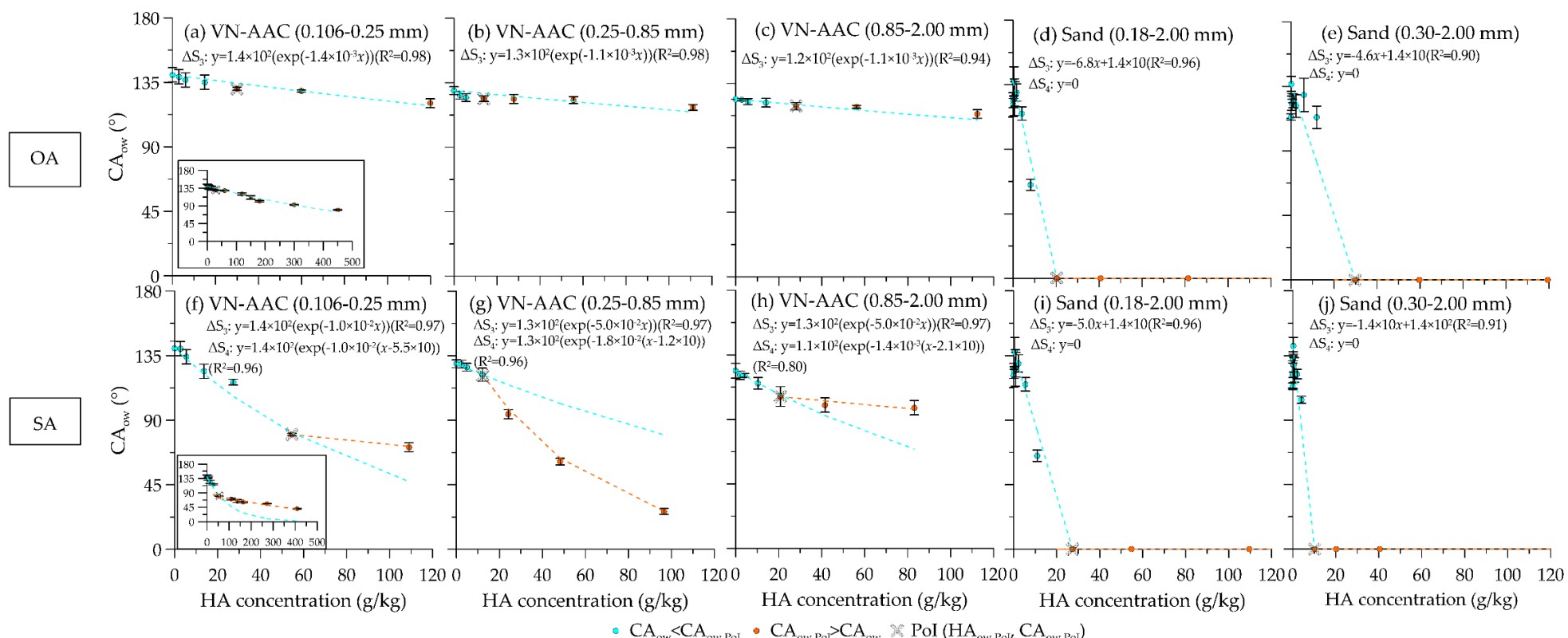

**Figure 5.** Relationships between $CA_{ow}$ and HA concentration of OA- and SA-coated VN-AAC grains and sands. OA-coated VN-AAC grains: (**a**) Grain size = 0.106–0.25 mm, (**b**) Grain size = 0.25–0.85 mm, and (**c**) Grain size = 0.85–2.00 mm. OA-coated sands: (**d**) Grain size = 0.18–2.00 mm and (**e**) Grain size = 0.30–2.00 mm. SA-coated VN-AAC grains: (**f**) Grain size = 0.106–0.25 mm, (**g**) Grain size = 0.25–0.85 mm, and (**h**) Grain size = 0.85–2.00 mm. SA-coated sands: (**i**) Grain size = 0.18–2.00 mm, and (**j**) Grain size = 0.30–2.00 mm.

As shown in Figure 5 and Table 2, the $CA_{ow,max}$ values of both OA- and SA-coated samples decreased with increasing grain size. The values of $CA_{ow,PoI}$ for SA-coated samples, however, were relatively constant (80–121°). The difference in the relationships between $CA_{ow}$ and HA concentration for OA- and SA-coated VN-AAC might be attributed to characteristics of the HA coating. To understand this, the measured $CA_{ow}$ were plotted against the measured specific surface area (SSA) and are shown in Figure 6. In the figure, the relationships between measured $CA_{wa}$ and SSA are also shown as a reference. Compared to the relationship between $CA_{wa}$ and SSA (Figure 6a,c,e), the relationships between $CA_{ow}$ and SSA showed a big difference between SA- and OA-coated samples, especially in the region of Step 2 (Figure 6b,d,f). For each grain sample of VN-AAC, the SSA of SA-coated samples was higher than those of OA-coated samples. This indicates that SA coated porous grains (AAC) effectively without reducing SSA and reduced the $CA_{ow}$ in Step 2 (oleophilicity) (Figure 5f–h) compared to OA-coated grains. In other words, the OA coating gave high $CA_{ow}$ (oleophobicity) even when controlling the HA concentration. Considering the concept of affinity of dispersed oil and oil trap adsorbents (trapping of dispersed oil in wastewater), high oleophilic adsorbents (i.e., high hydrophobic adsorbents) are preferable for the fixed-bed grains in the oil/water separation system. Thus, an SA-coated AAC grain is likely to be more effective for oil/water separation compared to an OA-coated AAC grain.

The relationship between measured $CA_{ow}$ as a function of HA concentration for sands, on the other hand, are shown in Figure 5d,e,j,i. The rapid linear decrease of $CA_{ow}$ can be observed with increasing HA concentration and became $CA_{ow} = 0°$ at HA= 9.4–28.9 g/Kg (corresponding to $HA_{ow,PoI}$; see also Table 2). The rapid decrease of $CA_{ow}$ means that coated OA and SA easily formed the grain surface oleophilicity, indicating that HA-coated sands are more suitable for oil/water separation than HA-coated AAC grains. As shown in Table 1, however, the SSA of sands are much smaller than those of AAC grains, implying that the contact area between dispersed oil in wastewater and the oleophilized grain surface area is decreased. Thus, further studies are needed to examine the efficiency of oil/water separation and trapping capacity of dispersed oil in wastewater by carrying out oil adsorption tests [65–68] and fixed-bed column experiments [69,70].

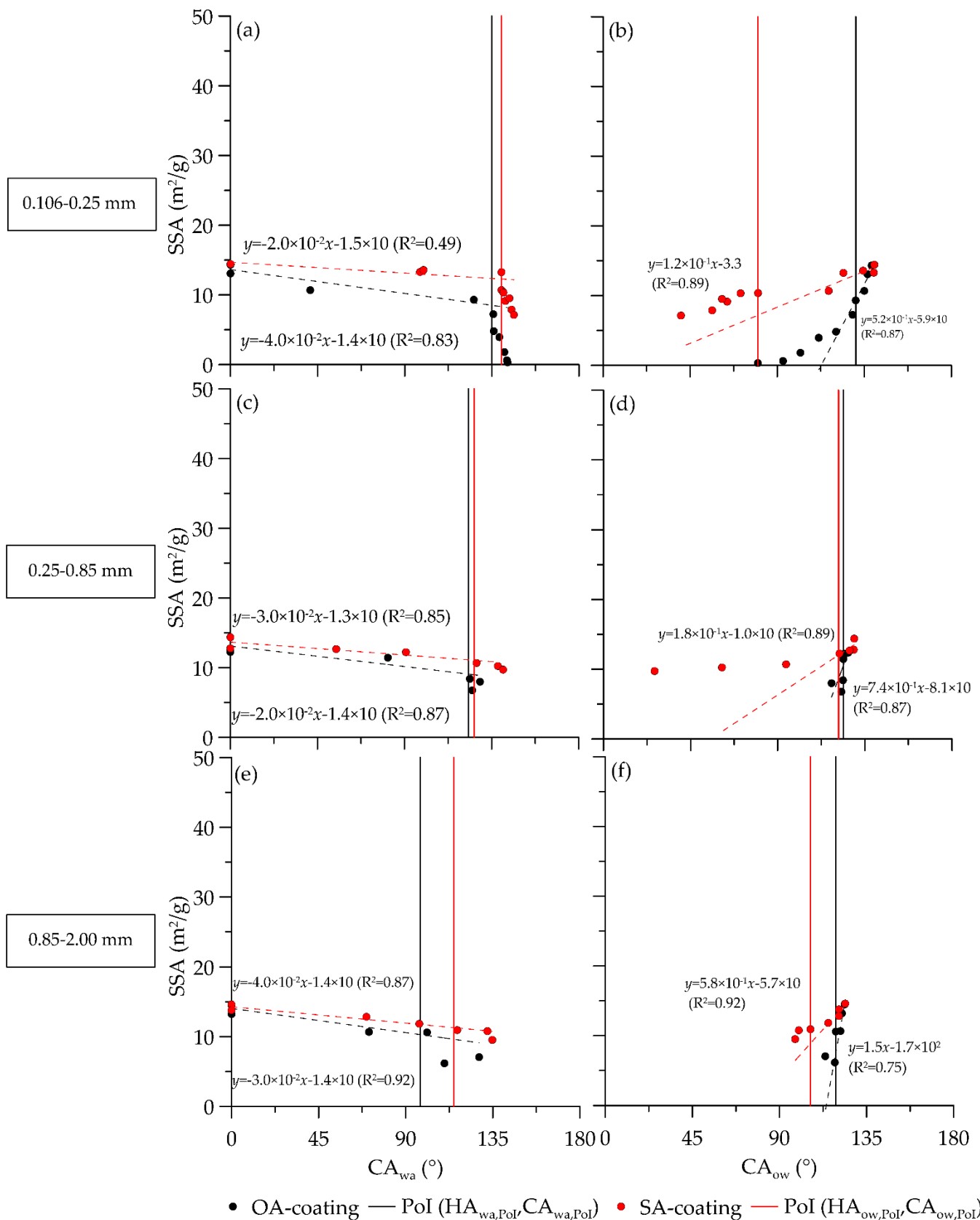

**Figure 6.** Relationships between SSA and $CA_{wa}$/$CA_{ow}$ for OA- and SA-coated VN-AAC grains. OA-coated VN-AAC grain: (**a**,**b**) Grain size = 0.106–0.25 mm, (**c**,**d**) Grain size = 0.25–0.85 mm, and (**e**,**f**) Grain size = 0.85–2.00 mm.

### 3.4. Relationships between Physicochemical Properties and Measured CA_wa and CA_ow

Finally, Pearson analysis was done to examine the correlations of measured $CA_{wa}$ and $CA_{ow}$ with physicochemical parameters in this study. The results are shown in Table 3. Note that data from the literature for sands [35,43,60] were also used in the analysis. The $|R^2|$ values > 0.8 are given in bold in the Table.

It can be clearly seen that the SSA correlated well with both $CA_{wa}$ and $CA_{ow}$ for VN-AAC grains ($|R^2| > 0.80$). The OC correlated well with $CA_{wa}$ and $CA_{ow}$ for OA-coated VN-AAC grains ($|R^2| = 0.72$ and $0.92$); however, it gave a low correlation with $CA_{wa}$ for SA-coated VN-AAC grains (($|R^2| = 0.50$). The A/B ratio calculated from FT-IR spectra, which are often used to characterize the hydrophobicity of soils [71–74], gave low correlations with $CA_{wa}$ for both VN-AAC grains and sands, suggesting the A/B ratio does not characterize the hydrophobicity (i.e., contact angle of water droplet in air) for tested cementitious AAC grains and sands.

**Table 3.** Pearson correlation matrix of measured parameters for VN-AAC grains and sands tested in this study. Values > |0.80| are given in bold.

| Sample | Coating | | CA_wa (°) | CA_ow (°) | SSA (m²/g) | OC (%) | A/B | No. of Samples |
|---|---|---|---|---|---|---|---|---|
| VN-AAC | OA | CA_wa (°) | 1.00 | −0.61 | **−0.88** | 0.72 | 0.72 | |
| | | CA_ow (°) | | 1.00 | **0.82** | **−0.92** | −0.49 | |
| | | SSA (m²/g) | | | 1.00 | **−0.91** | −0.69 | 25 |
| | | OC (%) | | | | 1.00 | 0.61 | |
| | | A/B | | | | | 1.00 | |
| | SA | CA_wa (°) | 1.00 | −0.66 | **−0.80** | 0.50 | 0.59 | |
| | | CA_ow (°) | | 1.00 | **0.86** | −0.75 | −0.75 | |
| | | SSA (m²/g) | | | 1.00 | **−0.81** | −0.74 | 25 |
| | | OC (%) | | | | 1.00 | 0.50 | |
| | | A/B | | | | | 1.00 | |
| Sand | OA | CA_wa (°) | 1.00 | −0.15 | −0.70 | −0.01 | 0.59 | |
| | | CA_ow (°) | | 1.00 | 0.75 | −0.78 | −0.61 | |
| | | SSA (m²/g) | | | 1.00 | −0.54 | **−0.82** | 23 |
| | | OC (%) | | | | 1.00 | 0.69 | |
| | | A/B | | | | | 1.00 | |
| | SA | CA_wa (°) | 1.00 | −0.55 | **−0.91** | 0.41 | 0.58 | |
| | | CA_ow (°) | | 1.00 | 0.66 | −0.75 | −0.31 | |
| | | SSA (m²/g) | | | 1.00 | −0.43 | **−0.85** | 24 |
| | | OC (%) | | | | 1.00 | 0.29 | |
| | | A/B | | | | | 1.00 | |

### 4. Conclusions

In order to develop low-cost and high-performance oil/water separating filtration media, this study assessed the hydrophobicity (oleophilicity) of porous grains made from AAC scrap coated with harmless hydrophobic agents (HA) such as oleic and stearic acids (OA and SA). For both OA- and SA-coated AAC grains, it was newly found the contact angles of water droplets in air ($CA_{wa}$) increased linearly with increasing HA concentration and then became almost constant and/or gently increased after a specific point (point of intersection, PoI). The HA concentration corresponding to PoI ($CA_{wa,PoI}$) can be applied to determine the minimum coating amount that gives the approximately maximum hydrophobicity of HA-coated porous grans as well as sands. The measured contact angles of an oil droplet in water ($CA_{ow}$) for AAC grains gave a gradual decrease exponentially with increasing HA concentration for both OA- and SA-coated AAC grains. Especially, the relationship between $CA_{ow}$ and HA concentration for SA-coated AAC became a

unique one, and the $CA_{ow}$ decreased with increasing HA concentration in two steps. Based on Pearson correlation analysis, both $CA_{wa}$ and $CA_{ow}$ for HA-coated AAC grains were well correlated with the specific surface area (SSA) of tested materials, indicating that the SSA is a good indicator to quickly assessment hydrophobicity/oleophilicity (i.e., $CA_{wa}$ and $CA_{ow}$) of HA-coated porous grains. Further studies are planned to quantitatively characterize oil/water separation efficiency using the HA-coated AAC grains in this study using the oil adsorption and fixed-bed column experiments.

**Author Contributions:** Conceptualization, A.M. and K.K.; Data curation, A.M. and K. K.; Formal analysis, A.M. and K.K.; Funding acquisition, K.K.; Investigation, A.M. and K.K.; Methodology, A.M. and K.K.; Project administration, K.K.; Resources, A.M. and K.K.; Software, A.M. and K.K.; Supervision, K.K.; Validation, K.K.; Visualization, A.M. and K.K.; Writing—original draft, A.M.; Writing—review & editing, A.M. and K.K. All authors have read and agreed to the published version of the manuscript.

**Funding:** This research was supported by the project of Japan Science and Technology Agency (JST), Japan International Cooperation Agency (JICA) on Science, and Technology Research Partnership for Sustainable Development (SATREPS) (no. JPMJSA1701).

**Institutional Review Board Statement:** Not applicable.

**Informed Consent Statement:** Not applicable.

**Data Availability Statement:** The data presented in this study are available on request from the corresponding author.

**Acknowledgments:** We thank the members of Hanoi University of Civil Engineering and Zafar Muhammad Junaid, a former student at Saitama University, for their support to the sample preparation in Vietnam and laboratory works.

**Conflicts of Interest:** The authors declare no conflict of interest.

## Abbreviations

The following abbreviations are used in this manuscript:

| | |
|---|---|
| AAC | Autoclaved aerated concrete |
| A/B | The ratio of hydrophobicity to hydrophilicity |
| BET | Brunauer Emmett Teller |
| CA | Contact angles |
| $CA_{ow}$ | Contact angles of oil droplets in water |
| $CA_{ow,max}$ | Maximum $CA_{ow}$ of in the measured range |
| $CA_{ow,PoI}$ | $CA_{ow}$ at PoI |
| $CA_{wa}$ | Contact angles of water droplets in air |
| $CA_{wa,max}$ | Maximum $CA_{wa}$ of in the measured range |
| $CA_{wa,PoI}$ | $CA_{wa}$ at PoI |
| $D_{10}$ | Grain size at 10% passing |
| $D_{50}$ | Grain size at 50% passing (Median diameter) |
| $D_{60}$ | Grain size at 60% passing |
| EC | Electrical conductivity |
| EDS | Energy dispersive X–ray spectroscope |
| $G_s$ | Specific gravity |
| HA | Hydrophobic agents |
| $HA_{ow,PoI}$ | HA concentrations corresponding to the PoI ($CA_{ow}$) |
| $HA_{wa,PoI}$ | HA concentrations corresponding to the PoI ($CA_{wa}$) |
| OA | Oleic acid |
| OC | Organic carbon |

| | |
|---|---|
| pH | Potential hydrogen |
| PoI | Point of intersection |
| PVDF | Polyvinylidene fluoride |
| SA | Stearic acid |
| SDM | Sessile droplet method |
| SEM | Scanning electron microscope |
| SSA | Specific surface area |
| $U_c$ | The Uniformity Coefficient |
| VN | Vietnam |
| VN-AAC | AAC scrap in Vietnam |
| WBP | Waste brick powder |

**Appendix A**

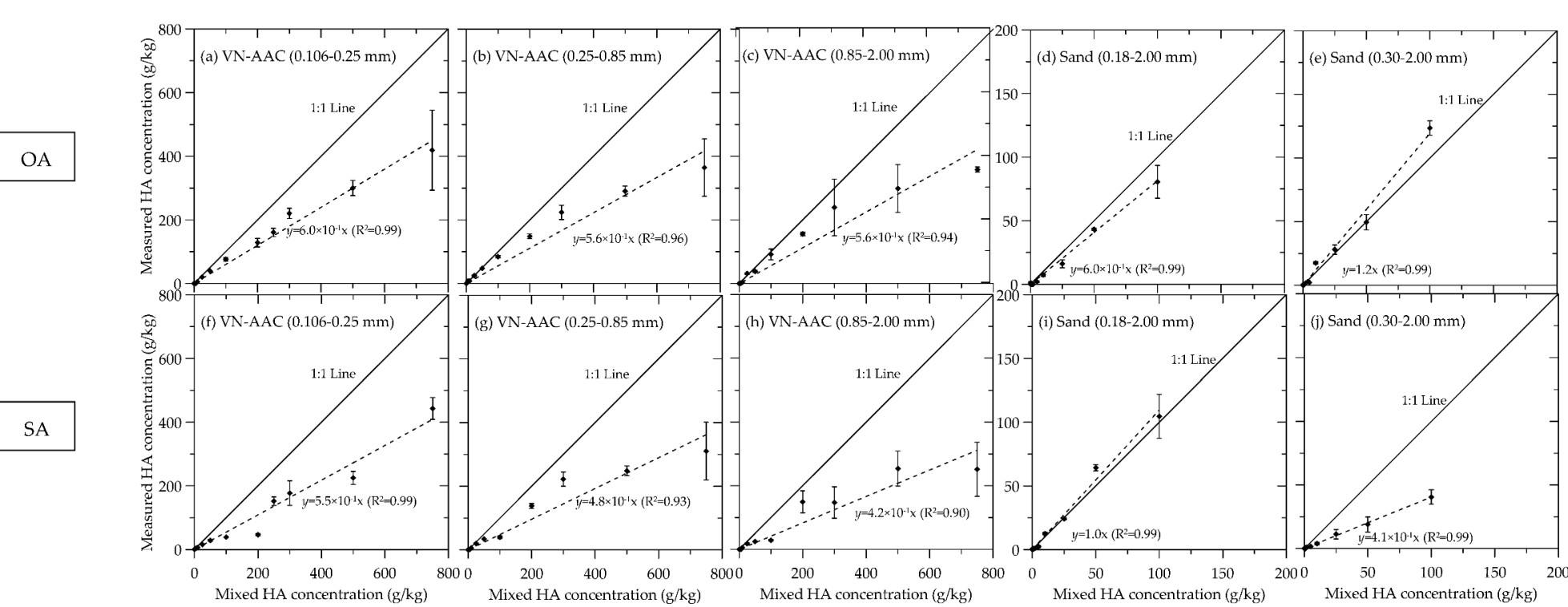

**Figure A1.** Relationships between measured and mixed HA concentrations for OA- and SA-coated VN AAC grains and sands. OA-coated VN-AAC grains: (**a**) Grain size = 0.106–0.25 mm, (**b**) Grain size = 0.25–0.85 mm, and (**c**) Grain size = 0.85–2.00 mm. OA-coated sands: (**d**) Grain size = 0.18–2.00 mm and (**e**) Grain size = 0.30–2.00 mm. SA-coated VN-AAC grains: (**f**) Grain size = 0.106–0.25 mm, (**g**) Grain size = 0.25–0.85 mm, and (**h**) Grain size = 0.85–2.00 mm. SA-coated sands: (**i**) Grain size = 0.18–2.00 mm and (**j**) Grain size = 0.30–2.00 mm.

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
