# Peer review of "Hydrophobicity/Oleophilicity of Autoclaved Aerated Concrete (AAC) Grains Coated with Oleic and Stearic Acids for Application as Oil/Water Separating Filtration and Adsorbent Materials in Vietnam"

_environments, doi:10.3390/environments9080101_

Round 1

Reviewer 1 Report

The manuscript entitled "Hydrophobicity/Oleophilicity of Autoclaved Aerated Concrete 2 (AAC) Grains Coated with Oleic and Stearic Acids for Application as Oil/Water Separating Filtration and Adsorbent Materials in Vietnam" includes a lot of work and contains many interesting data. I recommend its publication, with following remarks:  

- Introduction section is too long and must be compressed. The sentence on lines 37-43 is ambiguous and must be reformulated. The sentence on lines 54-57 requires references

- Materials and methods part contains informations like those on lines 116-120, 140-142  that should presented in Introduction 

- Results and discussion part: The authors  do not give enough insight into the presentation and discussion of results so as to support the research questions

Reviewer 2 Report

please find the separately attached reviewer's comments.
